# Antimicrobial, Antibiofilm, and Antioxidant Properties of Essential Oil of *Foeniculum vulgare* Mill. Leaves

**DOI:** 10.3390/plants11243573

**Published:** 2022-12-17

**Authors:** Michela Di Napoli, Giusy Castagliuolo, Natale Badalamenti, Viviana Maresca, Adriana Basile, Maurizio Bruno, Mario Varcamonti, Anna Zanfardino

**Affiliations:** 1Department of Biology, University of Naples Federico II, 80126 Naples, Italy; 2Department of Biological, Chemical and Pharmaceutical Sciences and Technologies (STEBICEF), Università degli Studi di Palermo, Viale delle Scienze, ed. 17, 90128 Palermo, Italy

**Keywords:** *F. vulgare* subsp. *vulgare* var. *vulgare*, Apiaceae, antimicrobial activity, antibiofilm property, antioxidant activity

## Abstract

*Foeniculum vulgare* (Apiaceae) is an aromatic fennel with important practices in medicinal and traditional fields, used in the treatment of digestive complications, and gastrointestinal and respiratory disorders. Its leaves and stems, tender and fresh, are used in the production of pasta dressing and main courses, while its seeds, with a strong smell of anise, are excellent flavoring for baked goods, meat dishes, fish, and alcoholic beverages. The aim of this work is concerning the extraction of essential oil (EO) from the leaves of *F. vulgare* subsp. *vulgare* var. *vulgare*, investigating antimicrobial, antibiofilm, and antioxidant efficacy. In particular, GC-MS analysis showed how the chemical composition of EO was influenced by the massive presence of monoterpene hydrocarbons (*α*-pinene 33.75%) and phenylpropanoids (estragole 25.06%). *F. vulgare* subsp. *vulgare* var. *vulgare* EO shows excellent antimicrobial activity against both Gram-positive and Gram-negative strains. This EO can inhibit biofilm formation at very low concentrations and has a good ability to scavenge oxygen radicals in vitro. *F. vulgare* subsp. *vulgare* var. *vulgare* EO also has an increased activity of superoxide dismutase (SOD), catalase (CAT) and glutathione peroxidase (GPx) enzymes and decreased ROS levels in zymosan opsonized PMNs (OZ).

## 1. Introduction

Apiaceae is a family of flowering plants, comprising 444 genera. This family has a wide distribution: from northern temperate regions to mountainous landscapes, up to tropical areas [1]. This geographical diversity is not accompanied by the structural diversity of plants. In fact, all the genera of this family are characterized by strong flavors and smells due to the presence of schizogonic ducts containing oil [2], mucilage, and resins, typical of both the aerial parts (leaves, stems, and fruits) and the roots [3]. The presence of different metabolites (coumarins, flavonoids, saponins, and terpenoids) allows the use of these plants in different sectors: food use (nutrition, drinks, and spices), pharmaceutical and cosmetic areas. Furthermore, many of them are used in traditional medicine for the treatment of gastrointestinal, reproductive, and respiratory diseases [2,4,5].

Moreover, from these plants it is possible to distil essential oils (EOs) with high yields and with great chemical variability [6,7]. Very different biological properties have been investigated and confirmed over the years; in addition to the antibacterial [8], antifungal [9,10], antioxidant [11], anti-inflammatory [12], and antitumor activities [13], their insecticidal potential has been very promising [14,15].

Belonging to the Apiaceae family, *F. vulgare* subsp. *vulgare* var. *vulgare* is a perennial with soft, feathery, almost hair-like foliage growing up to 6.6 ft (2 m) tall. This plant can look similar to dill. Leaves are striate, 3–4 pinnate, with segments filiform, up to 1.6 in (4 cm) long. The flowers are small, yellow, and found in large flat-topped umbels. Blooming occurs between July and October. Fruits from oblong to ovoid, 0.12–0.20 in (3–5 mm) long. Seeds ripen from September to October [16]. The dry fruit are used, also, in the culinary field. It is spontaneous species in the coastal regions of the Mediterranean Sea, but, by now, it has become widely naturalized in many parts of the world.

Furthermore, from our chemotaxonomic study [17], a clear distinction emerged, not only morphological, but also chemical, between two subspecies of fennel, *piperitum,* and *vulgare*. The subspecies *piperitum* has bitter seeds, while the *vulgare* subspecies has sweet seeds that are used as flavoring agents in baked goods, meat and fish dishes, alcoholic beverages, etc. for their characteristic aniseed smell [18].

The importance of this plant lies in the massive production, from the different plant parts, of the EO [19]. Extractable EOs, with a hint of anise, used in the culinary field [20], are generally characterized by the presence of phenylpropanoids such as (*E*)-anethole and estragole, and monoterpene hydrocarbons such as *α*-pinene and *α*-phellandrene) [18].

The EO obtained by hydrodistillation of the fruits showed clear antibacterial activity against *Escherichia coli*, *Bacillus megaterium* and *Staphylococcus aureus* [21], *E. coli* 0157:H7, *Listeria monocytogenes* and *S. aureus* [22,23], while the one extracted from the seeds had antimycobacterial and anticandidal activity [24]. However, it is also possible to underline excellent results as antioxidant [25], antidiabetic [26], and insecticidal agents [27].

Based on this promising background, the EO of the leaves of *F. vulgare* subsp. *vulgare* var. *vulgare*, obtained from a Sicilian population, has been tested and evaluated for potential antimicrobial, antibiofilm, and antioxidant activities. In particular, thanks to the high yields, the oil of *F. vulgare* subsp. *vulgare* var. *vulgare* has been tested as an antimicrobial agent against different Gram-positive and Gram-negative strains, showing excellent results. This EO was able to inhibit biofilm formation at very low concentrations (1–5 μg/mL) and had good oxygen radical scavenging ability in vitro. In addition, this EO has increased activity of superoxide dismutase (SOD), catalase (CAT), and glutathione peroxidase (GPx) enzymes and reduced levels of ROS in opsonized zymosan PMNs (OZ).

## 2. Results and Discussion

### 2.1. Chemical Profiling of F. vulgare subsp. vulgare var. vulgare EO

Hydrodistillation of the leaves of *F. vulgare* subsp. *vulgare* var. *vulgare* produced an intense yellow EO with a strong aniseed aroma. The chemical composition of this EO was previously described and reported by Ilardi et al. [17]. As shown in the graph of Figure 1, this mixture consisted mainly of monoterpene hydrocarbons (65.64%), with *α*-pinene (33.75%), *β*-pinene (5.13%), myrcene (5.25%), 3-carene (6.12%), and *γ*-terpinene-like (9.45%) main components.

The second most abundant chemical class was that of phenylpropanoids (30.36%), with the presence of estragole (25.06%) and (*E*)-anethole (5.30%). On the other hand, the oxygenated monoterpene compounds were present in minimal quantities (2.29%). The entire chemical composition is reported in Appendix A. Based on these observations, the antimicrobial, antibiofilm, and antioxidant potential of *F. vulgare* subsp. *vulgare* var. *vulgare* EO was explored.

### 2.2. Antimicrobial Activity of F. vulgare subsp. vulgare var. vulgare EO

*F. vulgare* subsp. *vulgare* var. *vulgare* has been used as an ethnic remedy for the cure of numerous infectious disorders of bacterial, fungal, viral, and mycobacterial origin. In the past, several studies have been carried out to assess its antimicrobial activity [24,28]. The compounds presented in the EO distilled from fennel leaves were used in different interesting studies [29,30,31,32]. To test so, we performed an inhibition halo assay, which provides a qualitative approach to understanding whether a particular compound possesses antibacterial activity. Figure 2 (panel 1A) displays a bacterial plate with the *E. coli* indicator strain, a model bacterium for Gram-negative, showing sensitivity to essential oil. Panel 1B of the same figure shows *S. aureus*, a model of a Gram-positive indicator strain, which also manifests sensitivity towards *F. vulgare* subsp. *vulgare* var. *vulgare* EO. The graph in Figure 2 (panel 2), having arbitrary units in mL, allows us to have a more precise idea of the antibacterial activity of our EO. The bigger the amount of EO used in the experiment, the greater the inhibition halo obtained, suggesting a proportionality between the quantity of used EO and its antimicrobial activity.

To learn more about the data on antibacterial activity, a viable cell count quantitativetype of test was performed. As shown in Figure 3 (panel A) the Gram-negative strains *E. coli, P. aeruginosa,* and *S.* Typhimurium manifest mortality against *F. vulgare* subsp. *vulgare* var. *vulgare* EO, *E. coli* being the most sensitive, having 100% mortality at the highest concentration (200 µg/mL). The same figure (panel B) shows the dose-response curves for Gram-positive bacteria: *S. aureus, M. smegmatis,* and *B. cereus*. *F. vulgare* subsp. *vulgare* var. *vulgare* EO is effective against all strains, causing 100% mortality at the maximum concentration (200 µg/mL) for both *S. aureus* and *B. cereus*. The antimicrobial activity of EOs is assigned to several small terpenoids and phenylpropanoids compounds, which, in pure form, also demonstrate high antibacterial activity [33].

We performed MIC experiments using the microdilution method. Table 1 shows the results. As expected, the lowest MIC values were calculated for *E. coli*, *S. aureus*, and *B. cereus* (250 µg/mL).

Through fluorescence microscopy, we tried to obtain some information about the *F. vulgare* subsp. *vulgare* var. *vulgare* EO mechanism of action, using *E. coli* and *S. aureus* indicator strains. The bacteria were treated with the EO at the maximum concentration used in the other tests, and two dyes were added: DAPI, a live cell DNA intercalator that gives blue coloration, and propidium iodide, a dead cell DNA intercalator that gives a red coloration through the damaged membrane. As shown in Figure 4, under optical microscopy conditions, the *E. coli* treated cells (panel C) show the same shape and color as the control (panel A). *E. coli* treated cells appear in blue (panel D), thus indicating no damage to the cell membranes, as well as those of the control (panel B). The same experiments on *S. aureus* confirmed the previously obtained results. Via optical microscopy, the treated cocci (Figure 4, panel G) are no different from the control ones (panel E), and the *S. aureus* cells appear in blue even after the treatment (panel H), similar to the control in fluorescence microscopy (panel F). Even for the Gram-positive strains, we can state that the bacterial membrane was intact after treatment with the EO. Essential oils and their components are known to be active against a wide variety of Gram-negative and Gram-positive bacteria. However, Gram-negative bacteria are more resistant to their antagonistic effects than Gram-positive ones, because of the lipopolysaccharide present in the outer membrane [34].

### 2.3. Antibiofilm Activity of F. vulgare subsp. vulgare var. vulgare EO

A crystal violet-based colorimetric assay was used to test the antibiofilm activity of *F. vulgare* subsp. *vulgare* var. *vulgare* EO. Figure 5 graph shows the *M. smagmatis* bacterial biofilm formation percentage, depending on the added oil concentration. The mycobacterium used is a non-pathogenic strain, a model for the microbial biofilms’ formation [35]. In this type of experiment, very low concentrations of EO were used (from 0 to 5 µg/mL), which have no effect on microbial growth, in such a way that the effect of reduction in the formation of the biofilm is linked only to the compound used and not to a decrease in cell vitality. The light gray curve in Figure 5 shows a good biofilm inhibition capacity (over 50%) at the highest concentration used. It is remarkable to use small quantities of a compound, in our case EO, to inhibit the formation of bacterial biofilms. This aspect has an essential impact on the potential use of the EO both in a natural environment to preserve the plants against pathogens [36] and in a medical one [34].

### 2.4. Antioxidant Activity of F. vulgare subsp. vulgare var. vulgare EO

The EO of *F. vulgare* subsp. *vulgare* var. *vulgare* is rich in hydrocarbon monoterpene, which has an antioxidant activity [37]. Figure 6 shows the increasing percentage of scavenging activities of ABTS and H_2_O_2_ radicals, as the concentration (1–1000 µg/mL) of EO increases. The data shown in Figure 6 are expressed in Table 2 as IC_50_ values, representing the EO concentration that causes a 50% reduction in ABTS and H_2_O_2_ radicals. The *F. vulgare* subsp. *vulgare* var. *vulgare* EO shows anti-H_2_O_2_ activity with IC_50_ values of 100 µg/mL and the lowest anti-radical effect (IC_50_ value > 100 µg/mL) for ABTS. Cell survival experiments were performed against HaCat cells (immortalized human keratinocytes) at different concentrations up to 250 µg/mL. By MTT assay after 24 and 48 h of incubation with the *F. vulgare* subsp. *vulgare* var. *vulgare* EO, the treated cells were comparable to the control ones. From these experiments we can conclude that the EO of *F. vulgare* subsp. *vulgare* var. *vulgare* is not toxic for this cell line under the experimental conditions used.

### 2.5. ROS Generation and Antioxidant Enzymes Activity on Polymorphonuclear Leukocytes (PMN)

The antioxidant activity was investigated by testing the EO extract of *F. vulgare* subsp. *vulgare* var. *vulgare* on OZ-stressed PMNs. Both ROS levels and the activity of SOD, CAT, and GPx enzymes were evaluated (Figure 7). Following the stress induced by OZ, there is a significant increase in ROS, but by treating PMN with EO, a gradual reduction proportional to the increase in concentration was observed. Indeed, already in the PMNs treated with 1 µg of EO, a significant reduction of the ROS levels was observed, and moreover, in the PMNs treated with 100 µg and 200 µg of EO, the ROS levels show levels comparable to the control (PMN not stressed). Regarding the activity of antioxidant enzymes in PMNs treated with EO, they show the same trend. Indeed, the activity of CAT, SOD, and GPx increases statistically with increasing EO concentration.

A decrease in ROS is probably due to the increased activity of antioxidant enzymes; in fact, in the ROS detoxification cascade, SODs are the first antioxidant defense enzymes, catalyzing the dismutation of superoxide anions. The H_2_O_2_ generated by, e.g., O_2_^−^ dismutation by SODs, is further detoxified by the action of catalases.

The observed antioxidant activity has correlated with the chemical composition of the EO. It seems plausible that their main constituents, such as estragole (25.06%), *γ*-terpinene (9.45%), and *α*-pinene (33.75%), may play a significant role in the antioxidant action of EO. This is confirmed by a different study on the antioxidant activity of several EO components, in which many of the compounds present in these EOs show antioxidant effectiveness [38].

## 3. Materials and Methods

### 3.1. Plant Material and Isolation of Essential Oil

Leaves on the stems of *F. vulgare* subsp. *vulgare* var. *vulgare* were collected in July 2020 in Rocca Busambra, Palermo (Italy) and identified by Prof. Vincenzo Ilardi (3750′51.60″ N; 1321′20.75″ E; 700 m a.s.l.). An herbarium sample is present in the Herbarium Mediterraneum Panormitanum, Palermo, Italy. Using the Clevenger’s apparatus, 120 g of fresh leaves were hydrodistilled according to the indications reported by the European Pharmacopoeia [39]. The EO, obtained with a yield equal to 0.68% (*v/w*), once dried with sodium sulphate, showing an intense yellow color, was stored in the freezer at 20 °C.

### 3.2. GC-MS Analysis

Analysis of EO was performed according to the procedure reported by Rigano et al. [40].

### 3.3. Bacterial Strains

Gram-negative strains *Escherichia coli* DH5α, *Pseudomonas aeruginosa* PAOI ATCC 15692, and *Salmonella* Typhimurium ATCC14028; and Gram-positive strains *Staphylococcus aureus* ATCC6538P, *Bacillus cereus* ATCC10987, and *Mycobacterium Smegmatis* mc^2^ 155, were used to evaluate antimicrobial activity.

### 3.4. Antimicrobial Activity Assay

The presence of antimicrobial molecules in EO of F. vulgare subsp. vulgare var. vulgare was detected using the method of Kirby-Bauer with modifications [41]. Three different volumes (1, 10, and 50 µL) of EO concentrated 22 mg/mL were placed on Luria bertani agar plates that were overlaid with 10 mL of soft agar (0.7%) and pre-mixed with 10 μL of E. coli DH5α and S. aureus ATCC6538P grown for 24 h at 37 °C. The negative control was 50 µL dimethylsulfoxide (DMSO) 80% used to resuspend the F. vulgare subsp. vulgare var. vulgare EO; the positive control was represented by the antibiotic ampicillin (1 µL) concentrated 22 mg/mL. Plates were incubated overnight at 37 °C and the antimicrobial activity was calculated according to the relation cited below [42].
(1)$$A/\mathrm{mL}=\frac{\mathrm{Diameter}\;\mathrm{clearance}\;\mathrm{zone}\;\left(\mathrm{mm}\right)\times 1000}{\mathrm{Volume}\;\mathrm{taken}\;\mathrm{in}\;\mathrm{the}\;\mathrm{well}\;\left(\mu L\right)}$$

Another method to evaluate the antimicrobial activity involved the cell viability counting of the Gram-positive and Gram-negative strains. Bacterial cells were incubated with both essential oils at 1, 10, 100, and 200 µg/mL concentration. Bacterial cells without essential oils represented the positive control and instead cells with DMSO at 80% were used as the negative control. The following day, the surviving percent of bacterial cells was estimated by counting the number of colonies [43]. Each experiment was carried out in triplicate and the reported result was an average of three independent experiments. (*p* value was <0.05).

### 3.5. Determination of Minimal Inhibitory Concentration

Minimal inhibitory concentrations (MICs) of *F. vulgare* subsp. *vulgare* var. *vulgare* EO against the Gram-positive and Gram-negative strains were determined according to the microdilution method established by the Clinical and Laboratory Standards Institute (CLSI). A total of ~5 × 10^5^ CFU/mL were added to 95 µL of Mueller-Hinton broth (CAM-HB; Difco) supplemented or not with various concentrations (1–250 µg/mL) of *F. vulgare* subsp. *vulgare* var. *vulgare* EO [44]. After overnight incubation at 37 °C, MIC_100_ values were determined as the lowest concentration responsible for no visible bacterial growth. Each experiment was performed in triplicate and the reported result was an average of three independent experiments.

### 3.6. Antibiofilm Activity Assay

Crystal violet dye was used to evaluate the biofilm formation of *M. Smegmatis* mc^2^ 155. A 24 wells plate was prepared in which each well contained a final volume of 1 mL; the negative control was represented by only bacterial cells and medium, the positive control was represented by bacterial cells with antibiotic kanamycin 2 µg/mL, the other samples contained cells and EO [1, 2.5, and 5 µg/mL]. The plate was incubated at 37 °C for 36 h. The OD of the crystal violet present in the distaining solution was measured at 570 nm by spectrophotometric reading, carried out with a Multiskan microplate reader (Thermo Electron Corporation, Waltham, MA, USA) [45]. The biofilm formation percentage was calculated by dividing the OD values of samples treated with EO and untreated samples.

### 3.7. DAPI/PI Dual Staining and Fluorescence Microscopy Image Acquisition

For dual staining, 100 µL of the bacterial culture of *E. coli* DH5α and *S. aureus* ATCC6538P (bacteria were grown to mid-logarithmic phase) was incubated in the dark for 2 h at 37 °C in agitation in the presence or absence of *F. vulgare* subsp. *vulgare* var. *vulgare* EO, a concentration of 200 µg/mL. After the incubation, 10 µL of bacterial culture was mixed with DAPI solution (40, 6-diamidino-2-phenylindole dihydrochloride; Sigma Aldrich, Milan, Italy) (1 µg/mL DAPI final concentration) and PI (propidium iodide; Sigma Aldrich, Milan, Italy) 20 µg/mL. Samples were observed using an Olympus BX51 fluorescence (Olympus, Tokyo, Japan) using a DAPI filter (excitation/emission: 358/461 nm) [46].

### 3.8. ABTS Scavenging Capacity Assay

This assay was performed according to the reported method [47], with some modifications, which are based on ABTS radical cation scavenging. Then, 1 mL ABTS solution was added to 100 µL of EO (1; 10; 100; 200, and 250 µg/mL concentrations). The absorbance was measured at 734 nm against a blank, and the percentage inhibition of ABTS radical was determined from the following equation: ABTS^•+^ radical scavenging activity (%) = (1 − AS/AC) × 100, where AC is the absorbance of the ABTS solution and AS is the absorbance of the sample at 734 nm. The concentration required for 50% inhibition was determined and represented as IC_50_. Each experiment was performed in triplicate and the reported result was an average of three independent experiments.

### 3.9. Hydrogen Peroxide Scavenging Assay

Quantitative determination of H_2_O_2_ scavenging activity was measured by the loss of absorbance at 240 nm, as previously described by Beers and Sizer [48]. Different concentrations of EO (1; 10; 100; 200, and 250 µg/mL) were incubated at 20 °C in 1 mL of hydrogen peroxide solution [50 mM Potassium Phosphate Buffer, pH 7.0; 0.036% (*w/w*) H_2_O_2_]. After 30 min, the hydrogen peroxide concentration was determined by measuring the absorbance at 240 nm. The percentage of peroxide removed was calculated as follows: peroxide removed (%) = (1 − AS/AC) × 100, where AC is the absorbance of 1 mL of hydrogen peroxide solution and AS is the absorbance of the sample at 240 nm.

### 3.10. Eukaryotic Cell Culture

HaCat (human keratinocytes) cells are spontaneously transformed aneuploid immortal keratinocyte cell line from adult human skin, widely used in scientific research [45,46]. These cells were maintained in Dulbecco Modified Eagle Medium (DMEM), supplemented with 10% fetal bovine serum and 1% penicillin-streptomycin. Cells were cultured at 37 °C in a humidified atmosphere of 5% CO_2_. The EO of *F. vulgare* subsp. *vulgare* var. *vulgare* was added in a complete growth medium for the cytotoxicity assay [49,50].

### 3.11. ROS Generation and Antioxidant Enzymes Activity on Polymorphonuclear Leukocytes (PMN)

Whole blood was obtained with informed consent from healthy volunteers. Six healthy fasting donors were subjected to peripheral blood sampling with K3EDTA vacutainers (Becton Dickinson, Plymouth, UK). The PMN were isolated following the protocol described by Harbeck et al. [51]. The isolated PMNs were measured in the presence or absence of various concentrations of EO of *F. vulgare* subsp. *vulgare* var. *vulgare*, without or with opsonized zymosan (OZ).

Dichlorofluorescein (DCF) assay was performed to quantify ROS generation according to Manna et al. [52]. The PMN were treated with EO of *F. vulgare* at different concentrations (1; 10; 100; 200; µg/mL) without or with OZ (500 µg/mL) for 6 h and then incubated with the non-polar and non-fluorescent 2′,7′-dichlorodihydrofluorescin diacetate (DCFH-DA), at 10 μM final concentration, for 15 min at 37 °C. The ROS quantity was monitored by fluorescence on a microplate reader. Results were expressed as fluorescence intensity.

A commercial kit (BioAssay System, San Diego, CA, USA) was used to determine superoxide dismutase (SOD), catalase (CAT), and glutathione peroxidase (GPx) enzymatic activity in PMN cells according to the manufacturer’s recommendations. The activity of enzymes was expressed as U/L [53]. The EO of *F. vulgare* subsp. *vulgare* var. *vulgare* was tested at the concentration of 1; 10; 100; 200; µg/mL. The experiments were performed in the presence and absence of OZ (500 µg/mL).

### 3.12. Statistical Analysis

The data were examined by one-way analysis of variance (ANOVA), followed by Tukey’s multiple comparison post-hoc test. In Figure 6 and Figure 7, values are presented as mean st. err; numbers not accompanied by the same letter are significantly different at a *p* value < 0.05.

## 4. Conclusions

Most of the pharmacological studies have been conducted using uncharacterized crude fennel extracts. It is difficult to reproduce their results or identify the bioactive compounds. Therefore, chemical standardization and bioactivity-driven identification of bioactive compounds is required. However, fennel’s extensive traditional use and proven pharmacological activities indicate that there is still immense scope for its chemical exploration. In this study, the oil of *F. vulgare* subsp. *vulgare* var. *vulgare* was characterized by the occurrence of a high amount of phenylpropanoids (30.36%), such as estragole (25.06%) and (*E*)-anethole (5.30%), and of the monoterpene hydrocarbon *α*-pinene (33.75%). Its antimicrobial, and antioxidant properties have been investigated and the foundations have been laid for future studies. We have discovered remarkable antibiofilm properties at very low concentrations, which may represent a pioneering study for the use of the essential oil of the aerial parts of fennel to avoid the formation of biofilms and combat a scientific and public health problem of great importance. They should focus on validating the mechanism of action responsible for the various beneficial effects, and on understanding which plant-based compounds are responsible for such effects. The information requested, when available, will enhance our knowledge and appreciation for the use of fennel in our daily diets. Furthermore, the result of such chemical studies could further expand its existing therapeutic potential.

## Figures and Tables

**Figure 1 plants-11-03573-f001:**
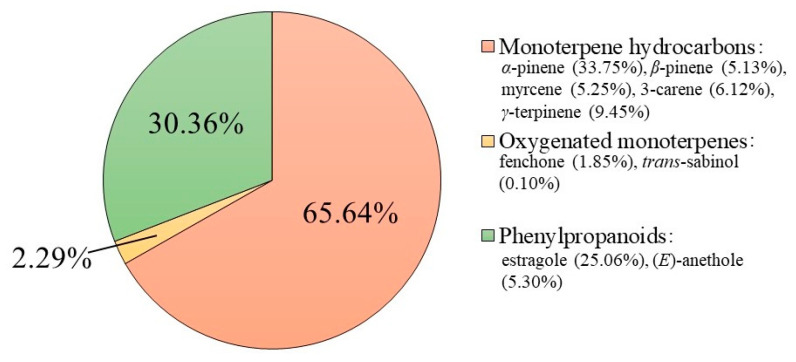
Chemical composition of EO obtained from the leaves of *F. vulgare* subsp. *vulgare* var. *vulgare*. The graph shows the total percentage of the single chemical classes and the majority of compounds identified by GC-MS.

**Figure 2 plants-11-03573-f002:**
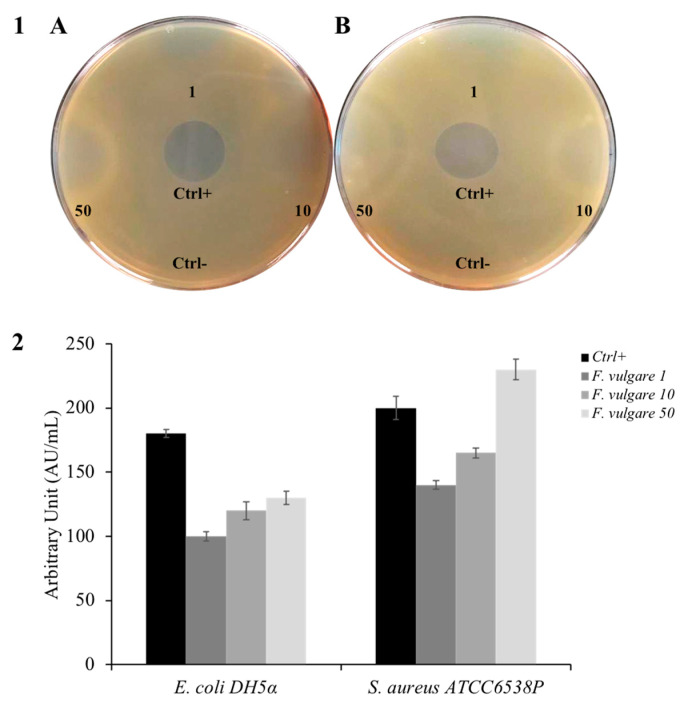
Kirby-Bauer assay. (**Panel 1**) shows the inhibition halo of *F. vulgare* subsp. *vulgare* var. *vulgare* against (**A**) *E. coli* and (**B**) *S. aureus.* (**Panel 2**) shows the inhibition halo expressed in AU/mL. Values are expressed as the average of three biological replicates; standard deviations are always less than 10%.

**Figure 3 plants-11-03573-f003:**
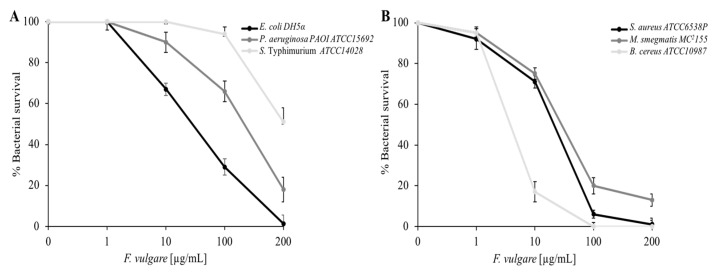
Determination of the *F. vulgare* subsp. *vulgare* var. *vulgare* EO antimicrobial activity at different concentrations evaluated by colony counting assay (**A**,**B**). The % Bacterial survival is represented on the y-axis obtained from the ratio of colony counts of treated and control. The assays were performed in three biological replicates; standard deviations are always less than 10%.

**Figure 4 plants-11-03573-f004:**
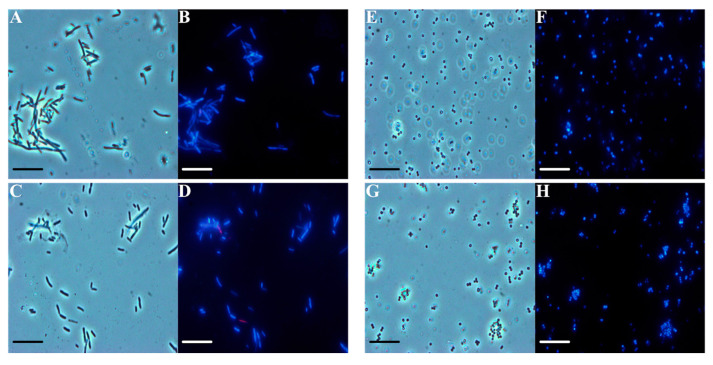
Evaluation of the antimicrobial action mechanism of *F. vulgare* subsp. *vulgare* var. *vulgare* EO, by fluorescence microscopy. Panels show *E. coli* bacterial cells (**A**–**D**) and *S. aureus* bacterial cells (**E**–**H**). Panels (**A**,**C**,**E**,**G**) show the cells observed under the optical microscope, and (**B**,**D**,**F**,**H**) under the fluorescence microscope. Untreated bacterial cells (**A**,**B**,**E**,**F**); cells treated with *F. vulgare* subsp. *vulgare* var. *vulgare* (**C**,**D**,**G**,**H**). Scale bars: 1 µm (**A**–**H**).

**Figure 5 plants-11-03573-f005:**
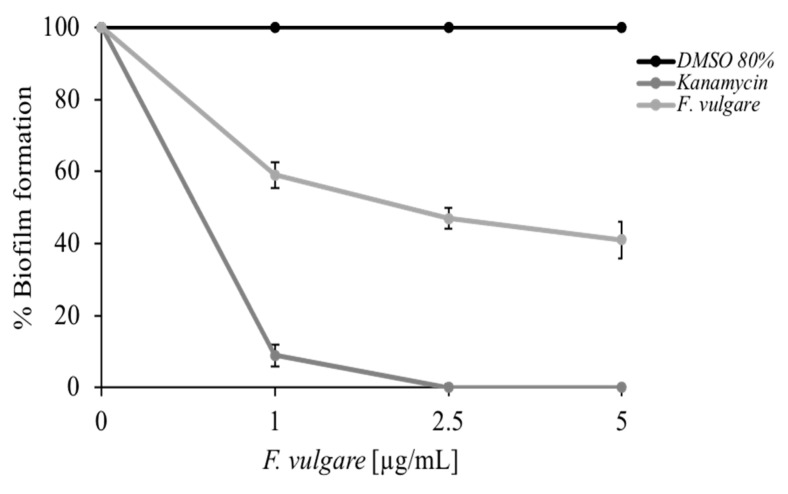
Colorimetric assay to evaluate the % of biofilm formation of *M. smegmatis*, at different concentrations of *F. vulgare* subsp. *vulgare* var. *vulgare* EO (1, 2.5, 5 μg/mL). The negative control is represented by DMSO (80%) and the positive control by Kanamycin. The assays were performed in three biological replicates; standard deviations are always less than 10%.

**Figure 6 plants-11-03573-f006:**
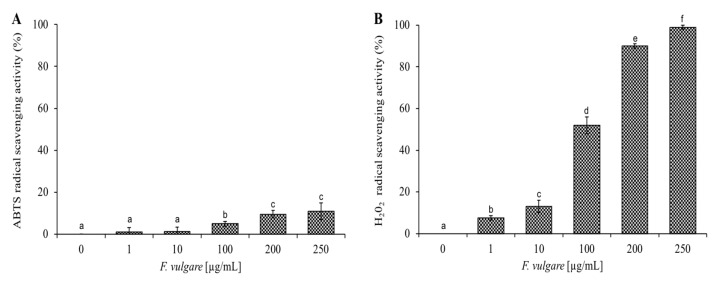
Determination of the antioxidant activity of *F. vulgare* subsp. *vulgare* var. *vulgare* EO. Panel (**A**) shows the abatement activity of ABTS radicals obtained after 10 min of incubation and reported as % of ABTS removed with respect to the control. Panel (**B**) shows the hydrogen peroxide scavenging activity, measured after 30 min of incubation, and reported as % of H_2_O_2_ removed relative to the control. Data were presented as mean and standard error and they were analyzed with a paired t-test. Bars not accompanied by the same letter were significantly different at *p* < 0.05.

**Figure 7 plants-11-03573-f007:**
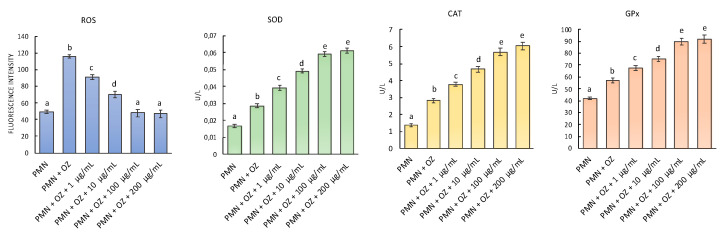
ROS generation and activities of antioxidant enzymes in PMN treated with EO of *F. vulgare* subsp. *vulgare* var. *vulgare* at the concentrations of 1, 10, 100, 200 µg/mL with or without OZ (500 µg/mL). Data were presented as mean and standard error and they were analyzed with a paired t-test. Bars not accompanied by the same letter were significantly different at *p* < 0.05.

**Table 1 plants-11-03573-t001:** Determination of the minimum inhibitory concentration values of bacterial growth (MIC_100_ expressed as µg/mL) of *F. vulgare* subsp. *vulgare* var. *vulgare* EO against Gram-negative and Gram-positive bacteria. The values were obtained from a minimum of three biological replicates.

Strains	MIC_100_ [µg/mL]
***E. coli* DH5α**	250
***P. aeruginosa* PAO1 ATCC15692**	>250
***S.* Typhimurium ATCC14028**	>250
***S. aureus* ATCC6538P**	250
***M. smegmatis* MC^2^155**	>250
***B. cereus* ATCC10987**	250

**Table 2 plants-11-03573-t002:** Representation of the IC_50_: Inhibiting Concentration Free Radical at 50%. ABTS: 2,20-azino-bis (3-ethyl-benzothiazoline-6-sulfonic acid); H_2_O_2_: hydrogen peroxide. The positive control is ascorbic acid for ABTS; and resveratrol for H_2_O_2_.

Sample	IC_50_ of ABTS (µg/mL)	Sample	IC_50_ of H_2_O_2_ (µg/mL)
*F. vulgare* subsp. *vulgare* var. *vulgare*	>1000	*F. vulgare* subsp. *vulgare* var. *vulgare*	100
Ascorbic acid	0.03	Resveratrol	0.05

## Data Availability

Samples of the compounds are available from the authors of the Department STEBICEF, University of Palermo, Italy.

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
