# Peer review of "Antimicrobial, Antibiofilm, and Antioxidant Properties of Essential Oil of Foeniculum vulgare Mill. Leaves"

_plants, 2022, doi:10.3390/plants11243573_

Round 1

Reviewer 1 Report

This paper by Di Napoli et al. reports the biological activity of the essential oils obtained from leaves of Foeniculum vulgare.

Foeniculum vulgare Mill. is a well known medicinal plant native to the Mediterranean area.

As correctly reported by the authors in the introduction section, the medical importance of this plant lies in the massive production, from the different plant parts, of the EOs that are, generally, characterized by the presence of  trans-anethole, estragole, fenchone, α-pinene and α-phellandrene.

Most importantly, the relative concentration of these compounds varies considerably depending on several factors  such  as  climatic  and  environmental  conditions,  harvesting season  and  the  stage  of  seed  ripening  can  influence  the  content  of essential oil and its composition in fennel.

Moreover, the chemical polymorphism of the essential oil composition depends upon the method of extraction.

Overall the findings are supported by data. However, the authors need to address some minor issues.

1.     In the results and discussion section, the authors describe (in 2.1 Chemical profiling of F. vulgare EO) the Chemical composition of EO obtained from the leaves of F. vulgare. In a previous study, the authors Badalamenti and Bruno have evaluated by GC and GC-MS the chemical compositions of the EOs from the leaves of Foeniculum vulgare subsp. piperitum, and compared them with those of the EOs of the same vegetative parts of Foeniculum vulgare subsp. vulgare, collected in the same station and in the same days in Sicily. The oils of F. vulgare subsp. vulgare showed completely different compositions, with estragole, (E)-anethole and α-pinene as main compounds, clearly indicating the differentiation of the two subspecies.

The authors should specify whether what is reported in figure 1 is a further analysis of the chemical profile of F. vulgare EO or is an analysis of data already published by Ilardi et al. If this is the case the authors should also remove the GC-MS analysis from the materials and methods section.

2.     “tiphymurium” and “thiphymurium” should be replaced by Typhimurium throughout the text and in Figures, and it should also not be in italic because Typhimurium is a serotype and not a species.

3.     In the materials and methods section the authors should specify:

-       the variety of Foeniculum vulgare used to extract essential oils is Foeniculum vulgare subsp. piperitum, instead of citing only the work by Ilardi et al.

-       the ATCC numbers of all bacterial strains used, (e.g. P. aeruginosa PAO1 ATCC BAA-47 or ATCC 15692 ?)

4.     The negative control was dimethylsulfoxide (DMSO) 80%

Authors should specify what they mean by 80% DMSO negative control. 

If 80% DMSO is used to resuspend F. vulgar EO, what is the final DMSO % in bacterial cultures? and so what is the % of DMSO in all negative controls?

The authors should edit the sentence in line 268 : cells with DMSO at 80% were used as the negative control.

5.     For the inhibition halo assay

It is not clear why in materials and methods the authors states that “different volumes of EO (22mg/ml)” were placed on agar plates, but in Figure 2 they reported ug of EO. In this type of modified assay (without paper filter) different microlitres cannot be used to compare the activity of different concentrations of the same substance (in this case EO).

Also, in Figure 2 it is not clear where is the negative control (80% DMSO, see also the previous consideration regarding % of DMSO of negative controls), nor the concentration of ampicillin used as a positive control. Please specify everything better in Materials and methods.

6.     In Figure 3 is reported the antimicrobial activity of EO at different concentrations (1, 10, 100, 200 μg/mL) evaluated by colony counting assay. % Bacterial survival is represented on the y-axis obtained from the ratio of colony counts of treated and control.

In the legend of figure 3, the authors should specify that is reported the ratio with the positive control. But in any case it is also necessary to report the % survival of the negative control (80% DMSO, see also the previous consideration regarding % of DMSO of negative controls).

7.     In materials and methods the authors describe the determination of MICs (MIC100 represents the lowest concentration responsible for no visible bacterial growth) by the microdilution method using different concentrations of EO (1–200 μg/mL), but in Table 1 MIC100 were  250 μg/mL. Why?

8.     For the Fluorescence Microscopy Image Acquisition, the authors should specify the object used or the scale bar in figure 4.

Author Response

Thanks for the precious comments. We have edited the manuscript in different parts as suggested.

Reviewer 2 Report

The examinations were carried out with essential oil extracted from fennel leaves. It would be important to indicate this fact in the title as well.

Latin names must be italicized, so Latin family names (Apiaceae, Umbelliferae) too.

Introduction chapter:

It is a little superficial.

The plant family Apiaceae does not need to be discussed in 2 paragraphs in the ‘Introduction’ chapter (it is too general), you should write about fennel right away.

The morphological characterization of the plant species is not scientific enough. The plant is perennial and not biennial.

The introduction of subspecies is very important. However, within ‘subsp. vulgare there are further subgroups: var. vulgare, var. dulce (see also the European Pharmacopoeia).

Reference should also be made to the European Pharmacopoeia and other official monographs.

In case of fennel twin achene fruits are mainly used, so information related to the herb and leaves of the plant in the literature would be very interesting and should be presented here.

Since the experiment was only carried out with essential oil, a long discussion of other active ingredients (e.g. phenolic compounds) is not interesting from the aspects of this topic.

On the other hand, it would be important to know what type and composition of fennel essential oils were used in the antimicrobial tests and with what results in the literature. If EO distilled from fennel leaves were also tested, it must be presented. If it has not been investigated before, this should be emphasized, because it gives the novelty and value of this research.

At the end of this paragraph the actuality and significance of the research should definitely be stated, why it is new, unique, and what is the practical or theoretical significance of it.

Materials and Methods chapter:

The plant material is not precisely defined, "Foeniculum vulgare" is not enough. It was not clear from the study that which subspecies were used for the EO distillation. If it was the subsp. vulgare, the varietas must also be specified (var. dulce or var. vulgare). The European pharmacopoeia also distinguishes between them.

The used plant parts should also be specified: in July, fennel is already in bloom. What kind of leaves were used for the essential oil distillation? Leaves of the leaf rosette, leaves on the stem or the whole flowering herb?

Results and Discussion chapter:

The exact composition of the EO included in the examination must be given, the pie chart is not professional enough. It should be indicate in table form with RT and with min. LRI values. Reference was made to Hardy et al. [16] in relation to the essential oil composition, but based on the title of this study, the authors investigated the subsp. piperitum. Was supsp. piperitum also applied here? This subspecies is not really used as a medicinal and spice plant.

When presenting the results, a methodological description is also included in the text in many places (e.g. lines 95-99, lines 136-140, etc.). Information about methods belongs to the Material and method subsection; only the presentation and evaluation of the results should be included in the ‘Results’ subsection.

Lines 120-121. "The antimicrobial activity of EOs is assigned to several small terpenoids and phenolic compounds, which, in pure form, also demonstrate high antibacterial activity [28]." – there are no phenolic components in the EO.

Line 84 - new topic, start in new paragraph.

Conclusion chapter:

The own results should be placed in the proper context here: what final conclusions were reached regarding the investigated properties of the essential oil of the given composition, what is the practical and theoretical significance of the results. Further research aims and directions can also be named here. But we need specifics here, practical results and conclusions!

Author Response

(The authors gave the same response as above.)

Reviewer 3 Report

The manuscript shows the biological properties of essential oils of Foeniculum vulgare, a common medicinal/herbal plant traditionally used in Asian countries. The authors are successful in demonstrating what they wish to do. Therefore, the manuscript is acceptable for publication.

Author Response

Thank you for reviewing our work.